# Failure Analysis of Topside Facilities on Oil/Gas Platforms in the Bohai Sea

**Songsong Yu [1], Dayong Zhang [2],\* and Qianjin Yue [2]**

[1] Faculty of Vehicle Engineering & Mechanics, Dalian University of Technology, Dalian 116024, China; 931679960@mail.dlut.edu.cn

[2] School of Ocean Science & Technology, Dalian University of Technology, Panjin 124221, China; yueqj@dlut.edu.cn

\* Correspondence: zhangdayong_2001@163.com

**Abstract:** The jacket platform in the Bohai Sea oilfield is an important engineering development, offering design alternatives in this economically important region. However, ice-induced vibration in cold areas threatens the safety and operation of these platforms. On two occasions, intense ice-induced vibrations triggered the rupture of a well's blow down pipeline, loosening the flanges on the Bohai platform, and leading to the ejection of high-pressure natural gas. Subsequent mechanical analysis of the failed parts helped define the mechanism of failure and identified the failure criteria, based on a prototype structure monitoring system. The analysis revealed that the deck's inertial force, which resulted in ice-induced steady-state vibration, was the major cause of the accident. Three fixed cones were thus installed on the platform the following winter, effectively reducing the vibrations.

**Keywords:** jacket platform; ice-induced vibration; failure mechanism; ice-resistant cone

## 1. Introduction

The shallow waters of the Bohai Sea are home to an abundance of oil reserves. More than 100 offshore platforms have been built on this gulf, making it the second-largest offshore oil production base in China. However, the Bohai Sea is a seasonal icing sea region and the ice is frequently driven by tides. Natural conditions, such as water depth, temperature, and presence of waves cause severe ice conditions in winter. Owing to the specific characteristics of marginal oilfields, the Bohai Sea oil and gas platform is an economically important jacket structure, to which the threat from sea ice is far greater than environmental hazards (wind, waves, currents, tides, earthquakes, etc.). Additionally, sea ice in cold regions poses higher risks to offshore oil platforms [1–3].

This paper discusses two serious accidents that took place on a Bohai platform, caused by intense ice-induced vibration. During the events, the unmanned wellhead platform, which was designed as a three-leg vertical jacket structure (Figure 1), was exposed to steady-state vibration that lasted for more than 10 minutes, resulting in the rupture of a well's blow down pipeline (Figure 2a). High-pressure natural gas was subsequently ejected, leading to automatic shutdown of the platform. An inspection of the site revealed a loose flange, which was the cause of the gas leak (Figure 2b). It was also made known that the currents occurred during slack tide, when ice speeds were slow. The thickness of the ice was found to be 8 cm and 11 cm, respectively, when the accidents occurred.

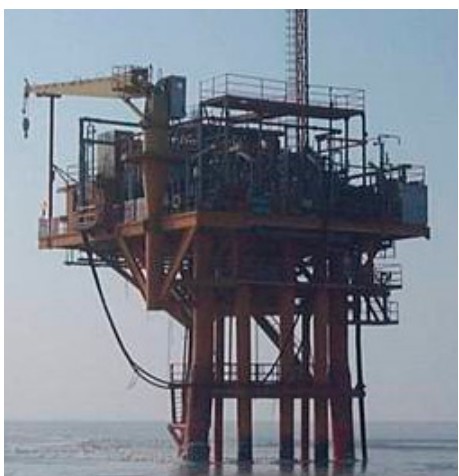

**Figure 1.** The natural gas platform where the accident took place.

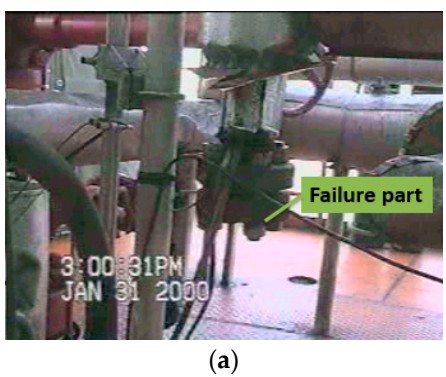

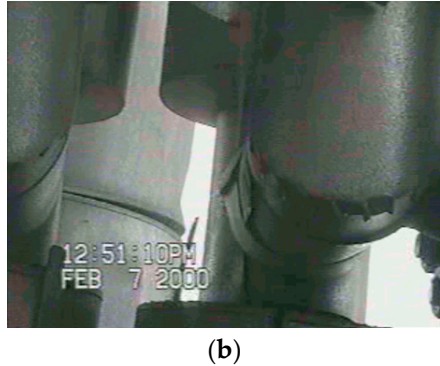

(**a**)  (**b**)

**Figure 2.** Sea-ice damage to platforms in Bohai: (**a**) Rupture of a blow down pipeline. (**b**) The loose flange.

This paper addresses the risk of oil and gas exploration in the Bohai Sea during winter. Based on data obtained by field monitoring systems, case studies were conducted on gas leakage accidents to illustrate the ice-induced failure of the oil and gas platform, and clarify the criteria of failure. It provides a theoretical basis to study the safety and security of oil and gas development in cold regions of Bohai Sea.

## 2. Failure Mechanism Analysis of the Upper Substructure of the Platform

### 2.1. Failure Analysis of Pipeline Fracture

Natural gas pipelines are an important component of offshore platforms. During a period of intense ice formation, the blow-down pipeline in the upper platform shook violently and was fractured. The ice-induced vibration resulted in the leakage of natural gas. In the vibration curve, obtained by field monitoring (Figure 3), it can be seen that there was a strong steady-state vibration of the platform when the accident occurred.

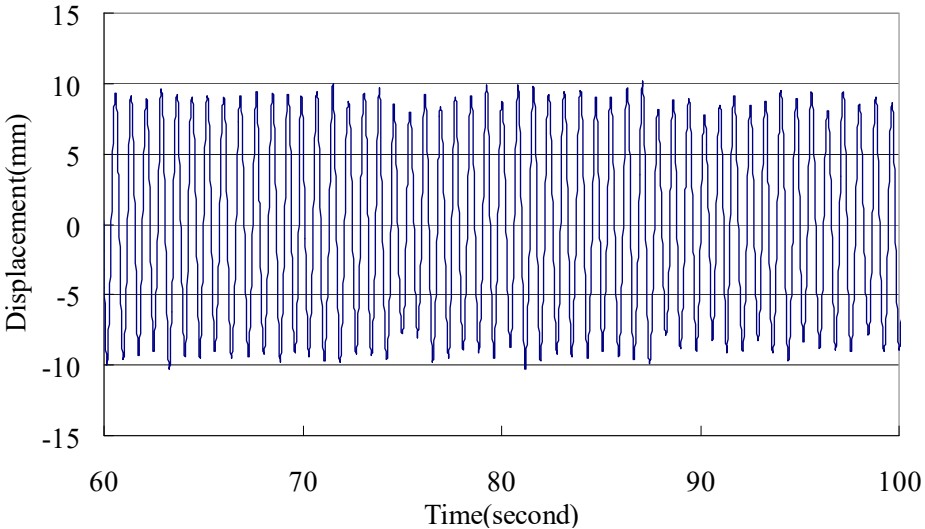

**Figure 3.** Displacement curve of the top deck when the pipeline fracture occurred.

Figure 4 shows a part of the blowdown pipeline after the accident. Mechanical analysis revealed that the blowdown pipe belongs to the cantilever structure and the fixed part is connected to the deck. The horizontal ice-induced shaking that occurred on the fixed part of the pipe was found to be equivalent to the bending vibration caused by a seismic load. This eventually caused a whiplash effect.

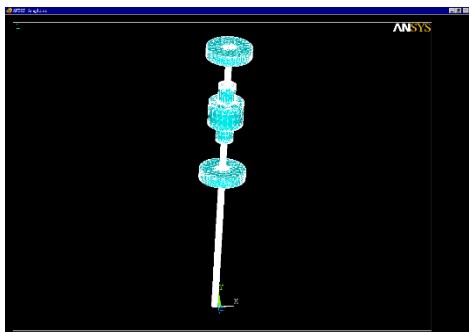

**Figure 4.** Partial structure of the blowdown pipe.

A fracture analysis of the affected section of the blowdown pipe (Figure 5) identified the following characteristics:

1. The fractured surface, which was a typical fatigue fracture, had an obvious fatigue source zone, a fatigue crack propagation zone, and a fatigue fracture zone.
2. The fracture was perpendicular to the axis, which meant that the fatigue fracture was caused by a bending load.
3. The fracture surface was uneven, with no obvious fatigue curve, and the distance between fatigue striations was large, illustrating low-cycle fatigue fracturing under high stress.

Study of the fixed part of the blowdown pipe indicates that the fatigue fracture load was a bending load. The following conclusions were made by analyzing the macro- and micro characteristics of the fracture: The fracture of the blowdown pipe had obvious fatigue features, and it exemplified a low-cycle fatigue fracture under a bending load. When the damage caused by alternating external loads reached a certain degree, it resulted in fatigue failure. The severe ice-induced vibration accelerated the fatigue failure of the blowdown pipe and eventually caused the ejection of high-pressure natural gas.

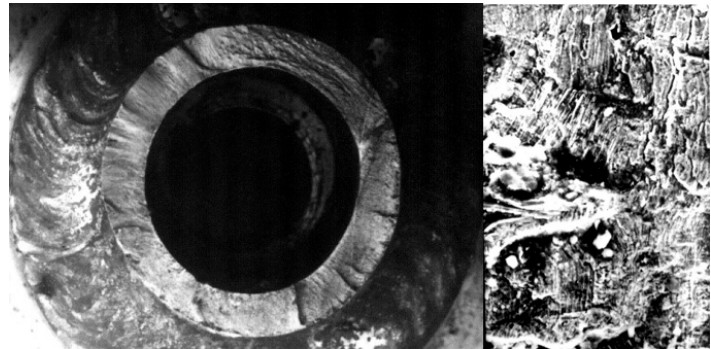

**Figure 5.** The fractured edge of the blowdown pipe.

### 2.2. Failure Analysis of Flange Looseness

Figure 6 shows the severe ice-induced vibration, which resulted in loosening of the flange of the platform's upper deck valve, and subsequent leakage of natural gas. An inspection of the platform after the accident revealed that the graphite spiral gasket flanges had 47% loose bolts, and the remaining had pretightening force of at least 50 N-m, after the system was fully depressurized. It was revealed that the reduction in flange pretightening force caused by bolts loose was the main reason for the flange failure.

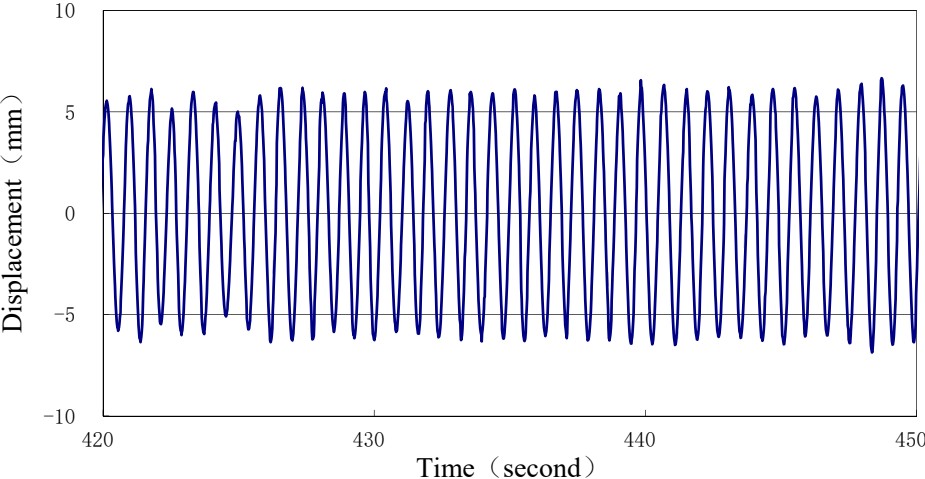

**Figure 6.** Displacement curve of the top deck when the accident occurred.

A flange is an important coupling part of industrial pipes, generally consisting of washers, flange bolts, and nuts. The flange's connection depends on the pretightening force between the bolt and the nut, transmitting pressure between the gasket and the flange; the washer undergoes elastic-plastic deformation to achieve a seal. It is liable to cause deteriorating connection and loose of the flange in the operating conditions where impact and vibration occur frequently. Several long-term studies have been conducted on the failure mechanism of vibration-induced bolt loosening. Goodier and Sweeney [4] conducted an experimental study on the loosening mechanism of bolted structures under an axial vibration load and found that self-loosening is caused by the relative movement of the thread and other contact surfaces. However, when a radial slip between the screw-thread occurs, the circumferential friction is zero. Because of the movement caused by normal contact stress, the screw–thread pair eventually slips in the circumferential direction. Gerhard Junker [5] studied the loosening mechanism of bolted structures under lateral loads and confirmed that transverse vibration was the main reason for it. Hess and Pai [6] from the University of South Florida used finite element analysis software to analyze contact state. They revealed that contact state can be divided into two types—local slip and complete slip—and cumulative local slips need much lower lateral loads. Jiang et al. [7,8] from the

University of Nevada studied the loosening of bolted structures under lateral vibration. In addition, research has shown that there is a stress limit for the threaded connection structure under lateral vibration. If this limit is adhered to, there will be no significant loosening on pressure from cyclic loads. If the bolt's pretightening force is increased, the loosening stress limit rises as well; however, this also increases the chances of fatigue fracturing of the bolt. Nassar and Housari [9] studied the bolt-loosening effect under various lateral vibration conditions. Studies have shown that the loosening torque and frictional force of the structure are affected by lateral vibration amplitude.

The inertial forces generated by the ice-induced vibration cause horizontal tremors in the topside facility, resulting in reduction of the pretightening force in the flange. When there is not much change in the static load and working temperature, the friction between the female and male screws keep the connection tight and satisfy the self-locking condition. The friction force between the spiral pairs instantaneously decreases or disappears when the platform experiences horizontal vibration. The threads on the bolt push into the thread surface of the nut due to the inclined surface, forcing the nut to rotate and release. Subsequently, when the pretightening force reaches a critical point, natural gas is free to escape. In order to further verify the looseness mechanism and process of the upper platform flange under ice-induced vibration, the study carried out a laboratory test of the system, based on results of the inspection performed after the accident.

It is difficult to simulate the pipeline flange situation in a laboratory, as the upper pipeline of the platform is complex and contains a long pipe. Considering that the vibrational load on the flange is a lateral one, the test intercepted a section of the flange and used the "four-point bending" model to simulate the situation. By using an actuator to control the amplitude and frequency, the bending stress at the flange and vibrational load on the bolt were revealed: both showed similar measurements to the actual situation, validating the rationality of the test.

Figure 7 shows the arrangement of the laboratory test: It comprised racks, a test pipeline flange, loading system actuator, and data acquisition system. The racks and flange form a beam structure that has concentrated forces at both fixed ends; the middle flange is stressed by bending. The amplitude and frequency of lateral alternating loads are controlled by actuators. Brackets and clamps are symmetrically applied to the pipeline on both sides of the flange, and the ends of the pipeline are fixed by a bolt and "v block" to the I-beam. This experiment simulated the accident's platform pipeline system under vibration using Class600 (11.0 MPa), 2-inch (DN50) nominal diameter, standard convexity welding neck flange, and 8 M16, high-strength bolts made of CrMoA. The metal–graphite, spiral-wound gasket consisted of a locating ring and an inner ring, and was made of 0Cr13.

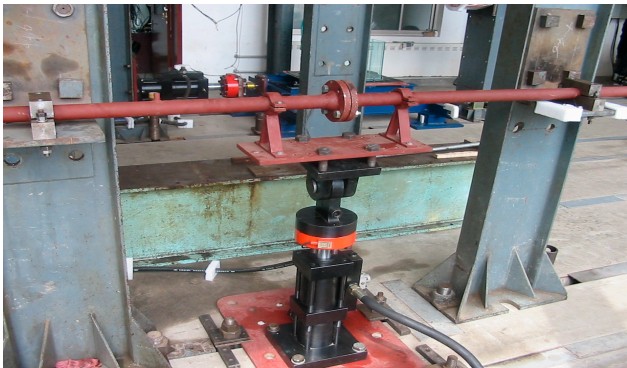

**Figure 7.** Arrangement of the laboratory test, which simulated the characteristics of the actual accident.

Based on the results of long-term monitoring and the statistics of pipe flanges used on the platform, the most common vibration acceleration levels under ice-induced vibration were found to be 1.0 m/s and 2.0 m/s, corresponding to two different ice conditions, respectively. Through vibration amplitude and frequency control, the bending stress and horizontal vibration acceleration simulation of the flange under ice-induced vibration was realized.

Numerical analysis has shown that experimental amplitude should be 4 mm for the experimental flange to be under real bending stress. In addition, when vibration frequency is set as 2 Hz, the vibration acceleration of the experimental pipeline is 1.0 m/s; when vibration frequency increases to 6 Hz, the vibration acceleration of the experimental pipeline is 2.0 m/s. The bolts are preloaded at 15%, 30%, and 45% of yield strength (640 MPa). Table 1 shows four groups of experimental vibrational parameters.

**Table 1.** Four groups of vibrational parameters.

| Initial Pretightening Force | Vibration Frequency | Amplitude | Acceleration | Time | Structure States | Final Pretightening Force |
|---|---|---|---|---|---|---|
| 18 KN | 2 Hz | 4 mm | 1.0 m/s$^2$ | 7 h | No | 17.5 KN |
| 18 KN | 6 Hz | 4 mm | 2.0 m/s$^2$ | 8 h | Yes | 190 N |
| 36 KN | 6 Hz | 4 mm | 2.0 m/s$^2$ | 7 h | No | 35.4 KN |
| 49 KN | 6 Hz | 4 mm | 2.0 m/s$^2$ | 17 h | No | 47.6 KN |

The first set of pretightening force is kept at 15% of the yield strength (18 KN), wherein the lower pretightening force can ensure that the graphite gasket is not crushed. Experimental data has found that when the excitation vibration acceleration is small (2 Hz), the pretightening force of the bolt hardly changes. In contrast, when the excitation frequency and acceleration are large, the bolt may loosen. When it occurs suddenly, the pretightening force is directly reduced from the initial value to almost zero (Figure 8).

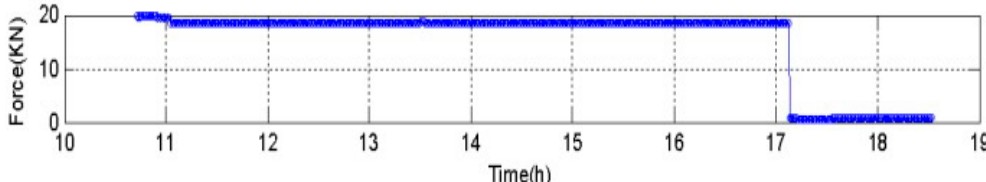

**Figure 8.** The pretightening force of the bolt when the initial pretightening force is 15% of the yield strength and excitation frequency is 6 Hz.

Figures 9 and 10 show the increase of the bolt's pretightening force to 30% and 45% of the yield strength, respectively. The analysis reveals that this force has a small decrease, even at a large vibration frequency. It is far from being loose, although the downward trend is obvious.

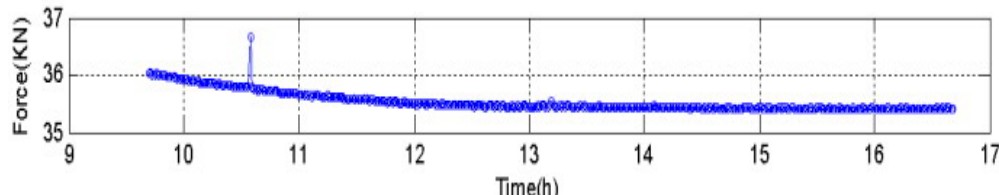

**Figure 9.** The pretightening force of the bolt when the initial force is 30% of the yield strength and the excitation frequency is 6 Hz.

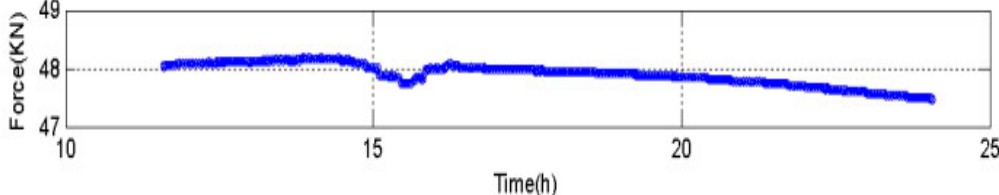

**Figure 10.** The pretightening force of the bolt when the initial force is 45% of the yield strength and the excitation frequency is 6 Hz.

Further tests on the flange in a vibrating environment show that a smaller initial pretightening force of the bolt can ensure a great sealing performance of the gaskets; however, the flange bolts will loosen under strong ice-induced vibration. Larger initial pretightening forces can delay or even prevent loosening, but excessive initial pretightening force may cause the gasket to yield prematurely, affecting its sealing performance.

## 3. Ice-Induced Steady-State Vibration Mechanism Analysis

Field monitoring has found that the periodic load generated by sea ice can evoke large acceleration responses in the platform. We performed spectral analysis of the above-mentioned offshore platform (Figure 11) using monitored response data. The results show that the vibration of the platform was focused on the first mode.

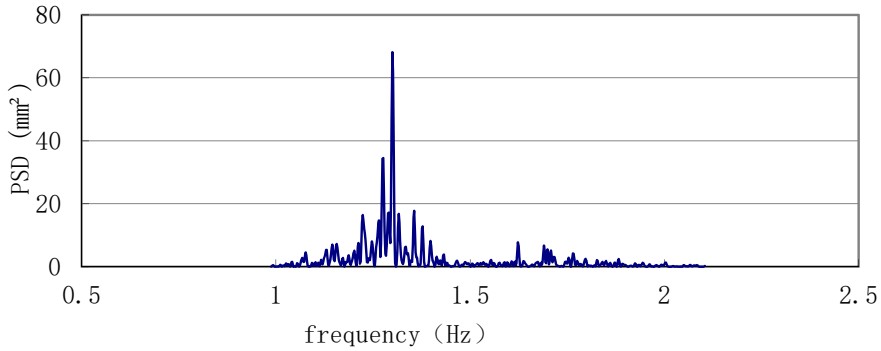

**Figure 11.** The studied offshore platform's main deck X-directional displacement spectrum.

The platform's vibration curve (Figures 3 and 6) shows that the tremor remained stable for a long time before the fracture of the blowdown pipe and gas leakage from the flange. The amplitude was consistently large and constant, about 30 mm, which means that a relatively strong steady-state vibration occurred in the early stage of the event. The frequency stabilized at 1.25 Hz, which is close to the platform's fundamental frequency, and the structure's vibration has obvious dynamic magnification.

Ice-induced vibrations of offshore structures pose a security challenge, and have been studied for decades in the regions of Cook Inlet, Beaufort Sea, and Baltic Sea. The steady-state vibration of the platform mentioned above is similar to the resonance in forced-vibration theory. The excitation frequency is close to the natural frequency of the structure, inducing dynamic magnification of the structural vibration. However, an analysis revealed that the change in frequency of the dynamic ice force depends upon the ice-breaking patterns and ice velocity. If such accidents are analyzed according to the forced-vibration theory, the frequency of dynamic ice forces will depend only on the ice. Thus, the steady-state alternating power is generated by sea ice, that is, it produces a breaking frequency. In fact, the mechanical property of natural sea ice has very large dispersion, and it is impossible to produce long-lasting stable sizes of crushing ice.

Blenkarn [10], Määttänen [11–13], and Engelbrektson [14,15] state that ice-induced steady-state vibration has typical self-excited characteristics. The steady-state vibrations of said platform occurred when the currents were in slack tide. Sea ice moves at a slow speed, taking into consideration the time currents take to change directions. Thus, the measured vibration curve analysis concludes that the structure and sea ice have similar speeds in a steady-state vibration period. The same direction and reverse movement of the structure and sea ice form a cyclic loading on the ice sheet and control its breaking period.

As a material, ice is very sensitive to the loading rate. Figure 12 shows that the uniaxial compressive strength of ice varies significantly with the strain rate, and the mechanical behavior of ice changes greatly when subjected to compressive loads. According to the different macroscopic mechanical behaviors of ice, the strain rate is divided into ductile region, ductile–brittle transitional

region, and brittle region. Yue et al. [16] pointed out that different loading rates of sea ice, caused by the relative velocities of the ice and the structure, led to different vibrational modes of the jacket platform. The mechanism of ice-induced self-excited vibration lies in the ductile–brittle transition region of the ice sheet.

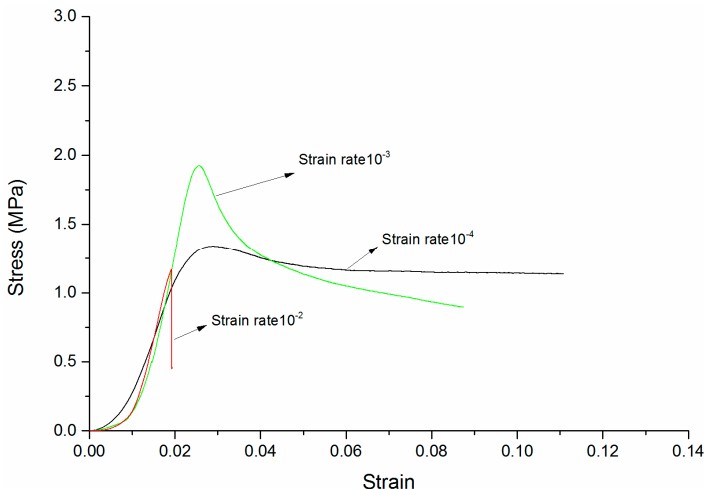

**Figure 12.** Sea ice load curves under different strain rates.

As shown in Figure 13, the cycle of self-excited vibration is divided into the loading phase, transition phase, and unloading phase. In the loading phase, the platform structure moves in the same direction as the sea ice at a similar speed. The loading speed of the ice sheet is slow, in the range of the ductile–brittle transition region. With loading of the ice force, new microcracks continue to be generated inside the ice, although they do not expand rapidly. As the loading rate continues to increase, the force is large enough to allow airfoil cracks to interpenetrate, and the ice sheet undergoes brittle failure. In the unloading stage, the direction of the structure's motion is opposite to that of the ice. The structure smashes the ice sheet, which was already damaged during the loading phase. It breaks down simultaneously, and the ice's force starts to unload. The structure swings back to the starting point of the loading phase, marking the end of a vibration cycle.

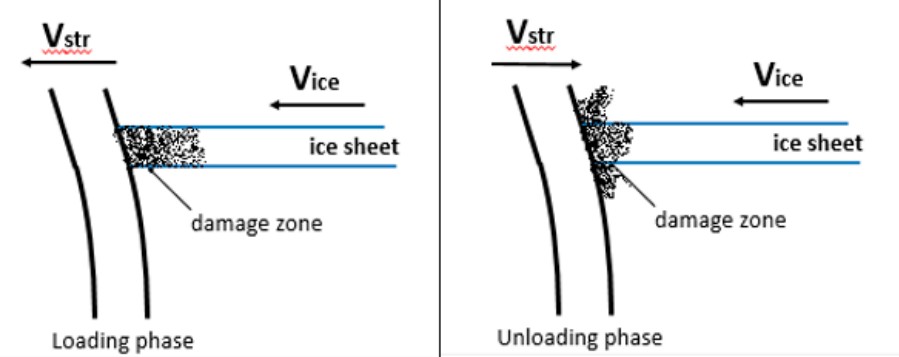

**Figure 13.** Ice-induced self-excited vibration physical mechanism.

The study revealed that since the loading rate of the ice sheet is in a ductile–brittle transition interval, ductile failure occurs on interaction between the ice sheet and the structure, and a large number of dislocation movements occur in the ice sheet [17]. With a relatively large compression deformation, contact between the ice sheet and the structure becomes more regular (Figure 14). The ice sheet ductile-to-brittle transition point marks the maximum uniaxial compressive strength corresponding to the loading rate (Figure 15) and the simultaneous breaking of sea ice, resulting from regular destruction, also aggravates the ice's force on the structure. That is the reason why the

breaking frequency is stable and the amplitude is extremely large during the steady-state vibration of the platform.

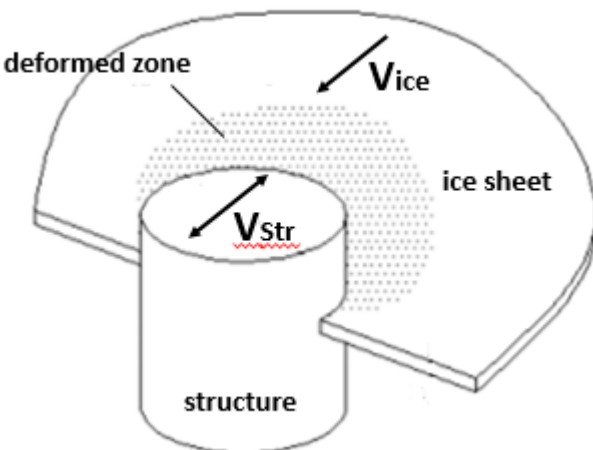

**Figure 14.** Outline of the ice sheet crushing the cylindrical structure.

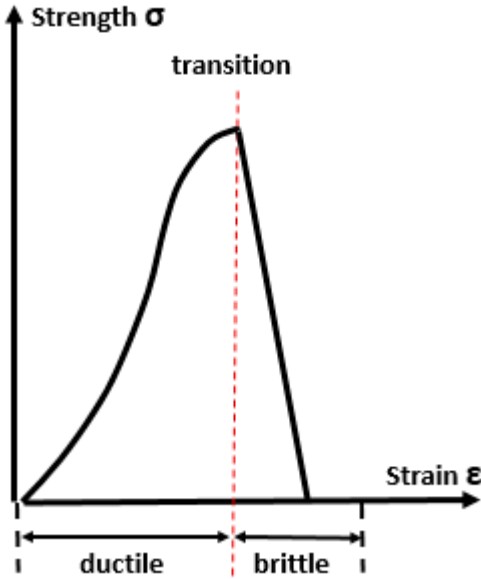

**Figure 15.** Loading curve of the ice sheet during strain rate increase.

The broken ice was able to easily bypass the platform's legs and avoid being stacked up due to the small diameter of the legs and the large amplitude and speed of the backswing, caused by the low rigidity of the platform. This results in synchronous crushing (loading and unloading processes) of the ice, and the ice's force also obtains a significant periodic variation in synchronization.

Ice-induced steady-state vibration on platforms is a major threat to safe oil and gas operations at Bohai Gulf. At a specific ice speed, ice extrusion produces a steady-state alternating excitation, which results in dynamic amplification of surface vibration. This further leads to fatigue and functional failure of the weak parts of the superstructure.

## 4. Evaluation of Ice-Resistant Modification

The conditions triggering the ice-induced self-excited vibrations are harsh. Although the frequency of this vibration is not high, it can generate a constant vibrational amplitude. Steady-state vibration is the main cause of ice-induced damage of marine structures, posing serious hazards.

The analysis discussed above resulted in an ice-resistant platform being created in the winter following the accident (Figure 16). By installing an ice-resistant cone, the ice sheet damage model changed from crushing destruction to bending destruction, thereby eliminating ice-induced self-excited vibration. In addition, continuous field-monitoring data was recorded, and the effect of the ice-resistant modification was evaluated.

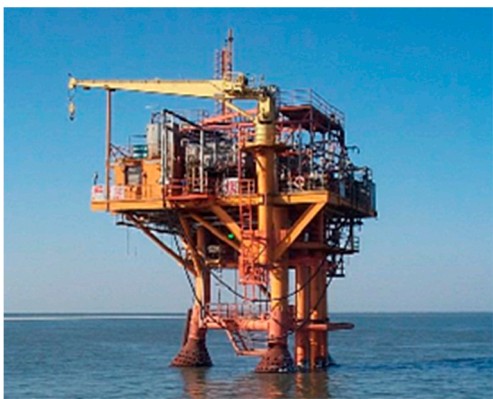

**Figure 16.** The ice-resistant modification of the platform.

Ice-induced failures in offshore platforms are always accompanied by steady-state vibrations. In order to highlight the variation in response characteristics of the platform structure after the modification, the 10-minute maximum amplitude data of the two years (before and after the accident) were compared. The average amplitude is presented in Table 2.

**Table 2.** Average maximum amplitudes (1999–2000).

| Vertical Structure | | Conical Structure | |
|---|---|---|---|
| Sample Starting Time | Average Maximum (mm) | Sample Starting Time | Average Maximum (mm) |
| Jan 28—22:57 | 8.38 | Jan 15—01:28 | 6.05 |
| Jan 28—23:07 | 5.64 | Jan 11—09:20 | 2.07 |
| Jan 29—00:27 | 12.2 | Jan 14—09:58 | 4.18 |
| Jan 29—00:37 | 13.9 | Jan 16—16:58 | 1.42 |
| Jan 17—07:51 | 1.88 | Jan 18—15:45 | 4.58 |
| Jan 24—04:42 | 4.32 | Jan 19—04:15 | 2.27 |
| Jan 27—08:48 | 4.78 | Jan 21—08:05 | 4.27 |
| Mean value | 7.3 | Mean value | 3.548 |

Liaodong Bay, which is one of the three bays in the Bohai Gulf, faced severe ice conditions in the winter following the ice-resistant modification. However, as a result of the installation of the cone, average peak values of the vibration significantly decreased, and a damping effect was obvious. With regard to the offshore platform's structural design, the modification caused a significant reduction in the average value of the peak vibration, thus vouching for its ice-resistant capability and fatigue life increase.

## 5. Conclusions

This paper addresses the ice-induced vibration risk of oil and gas exploration in the Bohai Sea through a case study of the pipeline and flange failure that occurred in Liaodong Bay. Based on the failure analysis of topside facilities on oil/gas platforms in the Bohai Sea, this paper recommends that the following criteria be considered for future design and risk assessment studies:

- Although the frequency of ice-induced, self-excited vibrations is not high, it is the main cause of ice-induced damage of marine structures, and must be paid more attention.
- Due to small leg diameter and low rigidity, the oil and gas platforms in Bohai are more likely to cause synchronous ice-crushing and ice-induced steady-state vibration in a range of ice speeds.

- The inertial forces generated by the ice-induced steady-state vibration may lead to fatigue and functional failure of weak topside facilities.
- The addition of ice-resistant cones results in a significant decrease of the average peak values of the vibration on the platform.

This research provides the theoretical basis for structural design and safety of offshore structures in cold regions. The authors recommend that further research, such as fatigue analyses, completing the evaluation of ice-induced vibration failure, and risk warning and forecasting be conducted on related problems.

**Author Contributions:** Project administration: Q.Y.; investigation and original draft writing: S.Y.; review and editing: D.Z.

**Funding:** This study was supported by the National Natural Science Foundation of China (Grant No. 51679033), National Key Research and Development Plan (Grant Nos. 2017YFF0210700, 2016YFC0303400), and Fundamental Research Funds for Dalian University of Technology (DUT18LK49).

**Acknowledgments:** The authors are pleased to acknowledge the support of this work by This study was supported by the National Natural Science Foundation of China (Grant No. 51679033), National Key Research and Development Plan (Grant Nos. 2017YFF0210700, 2016YFC0303400), and Fundamental Research Funds for Dalian University of Technology (DUT18LK49).

**Conflicts of Interest:** The authors declare no conflict of interest.

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
