# Peer review of "Failure Analysis of Topside Facilities on Oil/Gas Platforms in the Bohai Sea"

_jmse, doi:10.3390/jmse7040086_

Round 1

Reviewer 1 Report

This is an interesting paper.

Two minor corrections:

Line 86: Nm not nm

Line 135: 0Cr13, not oCr13.

Figure 3 - It appearsthat the data recording or processing is limited to approximately +/-14mm, so the max and min displacements are cropped and appear artificially uniform when they are outside +/-14mm. 

Author Response

Thank you for the reviewers’ comments concerning our manuscript entitled “Failure Analysis of Topside Facilities on Oil/Gas Platforms in the Bohai Sea”. The comments are helpful for revising and improving our paper. We have studied comments carefully and made correction which we hope meet with approval.

Point 1: Line 86: Nm not nm; Line 135: 0Cr13, not oCr13.

Response 1:

We are very sorry for our incorrect writing. We have made correction according to the Reviewer’s comments.

Line 82, “50n.m”were corrected as “50N.m”

Line 133, “oCr13”were corrected as “0Cr13”

Point 2: Figure 3 - It appears that the data recording or processing is limited to approximately +/-14mm, so the max and min displacements are cropped and appear artificially uniform when they are outside +/-14mm. 

Response 2:

We are very sorry for our negligence of using the defective curve.

Because of the excessive vibration, the displacement curve of the top deck is clipped without subjectivity manipulating. After the waveform recovering, the peak value is 32mm.

Line 53-54, We have replaced the defective curve with a displacement curve of the top deck at the same time period.

Figure 3. Displacement curve of the top deck when the pipeline fracture occurred.

We tried our best to improve the manuscript and made some changes in the manuscript. As the text has undergone English language editing by MDPI, we marked the changes in red in revised paper.

We appreciate for Reviewers’ warm work earnestly, and hope that the correction will meet with approval.

Once again, thank you very much for your comments and suggestions.

Reviewer 2 Report

the paper Is interesting and well structured

only minor revisions are required

In particular a deeper bibliographic analysis on methodologies to detect vibrations in this kind of facilities is suggested

Fig. 3 and Fig. 6  should be represented with the same axis scales

tables and figures should not be included in Conclusion paragraph

Author Response

Thank you for the reviewers’ comments concerning our manuscript entitled “Failure Analysis of Topside Facilities on Oil/Gas Platforms in the Bohai Sea”. The comments are helpful for revising and improving our paper. We have studied comments carefully and made correction which we hope meet with approval. Point 1: Fig. 3 and Fig. 6 should be represented with the same axis scales. Response 1: As Reviewer suggested, we redraw the Fig. 6 (Line 85) with same axis scales as Fig. 3 (Line 53). Figure 3. Displacement curve of the top deck when the pipeline fracture occurred. Figure 6. Displacement curve of the top deck when the accident occurred. Point 2: Tables and figures should not be included in Conclusion paragraph Response 2: Considering the Reviewer’s suggestion, I re-written this part under a section 4 and make Conclusion under a Section 5. In section 4, the ice-resistant modification effect is evaluated. Line 268-284, “Section 5” was added to recommend some criteria which should be considered in optimal design and risk assessment studies in the future. We tried our best to improve the manuscript and made some changes in the manuscript. As the text has undergone English language editing by MDPI, we marked the changes in red in revised paper. We appreciate for Reviewers’ warm work earnestly, and hope that the correction will meet with approval. Once again, thank you very much for your comments and suggestions.

Reviewer 3 Report

The results from the 4 point bending tests presented (at page 5-6 with Table 1 and Figures 8-10) show different structure states (for the bolt forces) at various pretightening forces and bending loads with excitatiton frequencies at 2 Hz and 6 Hz:

The operating conditions of the flanges at fullscale should be further elaborated: Descriptions/specifications of flange/bolts, pipe and operating conditions. If possible: Assumptions made for the tests which can serve to relate the test model to the flange/bolts at fullscale conditions should be highlighted.

Other causes (than the ice-induced vibration- eventually combined causes) to the detected flange looseness at fullscale were not discussed.

 L 251 "This paper addresses the risk of oil and gas exploration in the Bohai Sea" -is hardly correct

but: "This is a case  study of the pipeline and flange failure that occurred in Liaodong Bay." is more precise.

Platform failures always accompany steady-state vibrations. (always ?)

“In order to demonstrate the variation of the platform structure’s fatigue life after installing the ice-resistant cone, the ten-minute maximum amplitude data from the next winter after the ice-resistant modification were compared  with the previous year's. The average amplitude is shown in Table 2.”

The above section (/L 267-278) together with table 2  does not belong under Conclusions.

Suggestion:  Make the above section under a section 4 and make Conclusion under a Section 5.

(note: you do not demonstrate fatigue life variation but ten-minute maximum amplitude data)

Recommended to leave out or rare/unknown words: homodromous, heterodromous.

Details of Figure 4 and 5 are hardly readable in printed copy:

Figure 4 (better to use lower left part only?) and Figure 5 (resolution of text: lower right part)

Figure 12 lacks reference (and/or copyrights) and has low resolution in printed copy

Author Response

Thank you for the reviewers’ comments concerning our manuscript entitled “Failure Analysis of Topside Facilities on Oil/Gas Platforms in the Bohai Sea”. The comments are helpful for revising and improving our paper. We have studied comments carefully and made correction which we hope meet with approval. Point 1: The results from the 4 point bending tests presented (at page 5-6 with Table 1 and Figures 8-10) show different structure states (for the bolt forces) at various pretightening forces and bending loads with excitatiton frequencies at 2 Hz and 6 Hz: The operating conditions of the flanges at fullscale should be further elaborated: Descriptions/specifications of flange/bolts, pipe and operating conditions. If possible: Assumptions made for the tests which can serve to relate the test model to the flange/bolts at fullscale conditions should be highlighted. Other causes (than the ice-induced vibration- eventually combined causes) to the detected flange looseness at full scale were not discussed. Response 1: We have re-written this part according to the Reviewer’s suggestion to make the failure mechanism analysis more logical (Line 79-118).  Based on the monitoring ice-induced vibration curve of the top deck and subsequent maintenance, it was found that flanges had 47% of their bolts loose. It can be seen that the reduction in flange pre-tightening force caused by bolts loose is the main reason for the flange-loosening failure.  In light of the failure mechanism of a bolt-link loosening in operating conditions where impact and vibration occur frequently, long-term research shows that the horizontal vibration have been major contributors.  In order to further verify the looseness mechanism and process of the upper platform flange under ice-induced vibration, a laboratory test of the system was carried out, based on the monitoring results. As Reviewer suggested that operating conditions of the flanges at fullscale, assumptions and parameters selecting of the test was added (Line 136-145). Through the vibration amplitude and frequency control, the bending stress and horizontal vibration acceleration simulation of the flange under the ice-induced vibration is realized. And the bending stress and acceleration is Based on the long-term monitoring and statistics of pipeline flanges on the platform. Point 2: “In order to demonstrate the variation of the platform structure’s fatigue life after installing the ice-resistant cone, the ten-minute maximum amplitude data from the next winter after the ice-resistant modification were compared with the previous year's. The average amplitude is shown in Table 2.” The above section (/L 267-278) together with table 2 does not belong under Conclusions. Suggestion: Make the above section under a section 4 and make Conclusion under a Section 5. (note: you do not demonstrate fatigue life variation but ten-minute maximum amplitude data) Response 2: Considering the Reviewer’s suggestion, I re-written this part under a section 4 and make Conclusion under a Section 5. In section 4, the ice-resistant modification effect is evaluated. Line 256-258, the statements of “In order to demonstrate the variation of the platform structure’s fatigue life after installing the ice-resistant cone” were corrected as “In order to highlight the variation in response characteristics of the platform structure after the modification” Line 268-284, “Section 5” was added to recommend some criteria which should be considered in optimal design and risk assessment studies in the future. Point 3: L 251 "This paper addresses the risk of oil and gas exploration in the Bohai Sea" -is hardly correct, but: "This is a case study of the pipeline and flange failure that occurred in Liaodong Bay." is more precise. Response 3: Line 269-270, the statements of “This paper addresses the risk of oil and gas exploration in the Bohai Sea in winter through a case study of the pipeline and flange failure that occurred in Liaodong Bay.” were corrected as “This paper addresses the ice-induced vibration risk of oil and gas exploration in the Bohai Sea through a case study of the pipeline and flange failure that occurred in Liaodong Bay.” Point 4: Platform failures always accompany steady-state vibrations. (always ?) Response 4: Line 256, the statements of “Platform failures always accompany steady-state vibrations” were corrected as “Ice-induced failures in offshore platforms are always accompanied by steady-state vibrations.” Point 5: Recommended to leave out or rare/unknown words: homodromous, heterodromous. Response 5: Considering the Reviewer’s suggestion, I re-written this part under a section 5. And “homodromous \ heterodromous” was deleted. Point 6: Details of Figure 4 and 5 are hardly readable in printed copy: Figure 4 (better to use lower left part only?) and Figure 5 (resolution of text: lower right part) Figure 12 lacks reference (and/or copyrights) and has low resolution in printed copy Response 6: Line 59-62, we have replaced the Figure 4 and Figure 5 with clearer pictures. Figure 4. Partial structure of the blow-down pipe. Figure 5. The fractured edge of the blow-down pipe. Line 210-211, we have replaced the Figure 12, and the note of “Uniaxial compression strength vs. strain rate” were corrected as “Sea ice load curves under different strain rate”. Figure 12.Sea ice load curves under different strain rate. We tried our best to improve the manuscript and made some changes in the manuscript. As the text has undergone English language editing by MDPI, we marked the changes in red in revised paper. We appreciate for Reviewers’ warm work earnestly, and hope that the correction will meet with approval. Once again, thank you very much for your comments and suggestions.

Round 2

Reviewer 3 Report

    0Cr13; I assume OCr13 is the proper writing.